# “Pandemic Fatigue” in South America: A Multi-Center Report from Argentina, Bolivia, Paraguay, Peru, and Uruguay

**DOI:** 10.3390/brainsci13030444

**Published:** 2023-03-04

**Authors:** Julio Torales, Israel González-Urbieta, Iván Barrios, Marcela Waisman-Campos, Alexandra Terrazas-Landivar, Laura Viola, Tomás Caycho-Rodríguez, Osvaldo Melgarejo, Rodrigo Navarro, Oscar García, José Almirón-Santacruz, João Mauricio Castaldelli-Maia, Antonio Ventriglio

**Affiliations:** 1Department of Medical Psychology, School of Medical Sciences, National University of Asunción, San Lorenzo 001511, Paraguay; 2Department of Psychiatry, School of Medical Sciences, National University of Asunción, San Lorenzo 001511, Paraguay; 3Unit of Psychiatry, North East London NHS Foundation Trust, London RM13 8GQ, UK; 4Department of Statistics, School of Medical Sciences, National University of Asunción, Santa Rosa del Aguaray Campus, Santa Rosa del Aguaray 001511, Paraguay; 5Department of Neuropsychiatry, Fleni, Buenos Aires 2325, Argentina; 6Department of Neuropsychiatry, Universidad del Salvador, Buenos Aires 1699, Argentina; 7Department of Psychiatry, Mental Health Center, Universidad Domingo Savio, Santa Cruz de la Sierra 0701, Bolivia; 8Department of Child Psychiatry, Asociación Española, Montevideo 11600, Uruguay; 9School of Psychology, Universidad Científica del Sur, Lima 15067, Peru; 10Department of Neuroscience, Fundação do ABC, Santo André 19802-300, SP, Brazil; 11Department of Psychiatry, University of São Paulo, São Paulo 13566-590, SP, Brazil; 12Department of Clinical and Experimental Medicine, University of Foggia, 71122 Foggia, Italy

**Keywords:** COVID-19, pandemic, fatigue, pandemic fatigue, vaccination, South America

## Abstract

The COVID-19 pandemic has had a heavy impact on daily life, leading to physical and psychosocial consequences. Nowadays, clinicians and health researchers are particularly interested in describing and facing the long-term effects of COVID-19, also known as “long-COVID syndrome”. Pandemic fatigue has been defined as a cluster of demotivation, tiredness, and psychological effects that emerge gradually over time after the infection or through the adoption of the recommended measures to combat it. In this study, we report the findings of a large survey conducted in South America involving 1448 participants (mean age: 33.9 ± 11.2 years old) from Argentina, Bolivia, Uruguay, Peru, and Paraguay. An online survey was launched through the common social media based on a specific assessment aimed to detect the prevalence of pandemic fatigue and associated factors. Socio-demographic characteristics, medical, and personal information were collected; the Pandemic Fatigue Scale (PFS) and the Coronavirus Anxiety Scale (CAS) were also administered. We found mid-levels of pandemic fatigue among respondents (21.7 ± 7.95 score at PFS) as well as significant anxiety related to the COVID-19 pandemic (1.56 ± 2.76 score at CAS). In addition, pandemic fatigue was significantly associated with the experience of the loss of a relative/friend due to COVID-19, anxiety related to the infection, and reliance on social media as a primary source of information on the pandemic. Vaccination significantly reduced the levels of fatigue among respondents. Our findings may add to the international debate regarding the long-term health consequences of the COVID-19 pandemic and strategies to manage them in the general population of South America.

## 1. Introduction

The COVID-19 pandemic has brought unprecedented changes to daily life, affecting nearly every aspect of society in a sustained fashion for a considerable period of time [1]. On 22 May 2020, the World Health Organization (WHO) designated South America as the epicenter of the pandemic due to the high rate of COVID-19 cases in Brazil [2]. With some notable exceptions, most of the countries in the region adopted strong measures to delay the spread of the virus by implementing mass quarantines, travel bans, masking, and isolation requirements [3]. The pandemic became, for a long time, the focus of information providers, including traditional news services and social media. It dominated both the public scenes and private conversations.

The initial wave of research was rightly focused on the physical health impacts of the viral infection and gave way to the unprecedentedly rapid development of effective vaccines and life-preserving treatments. Nowadays, there is a growing interest in describing the long-term consequences of both the disease and measures taken to contrast it. Clinicians and health researchers are both facing the rise of the long-term effects of COVID-19, also known as “long-COVID syndrome”, including psychological effects [4,5,6].

The psychological effects of COVID-19 on the general population are likely multifaceted and may include the perception of personal vulnerability to the infection as well as worries about loved ones [7]. Quarantines as well as individual and societal restrictions have led to a range of negative emotional states such as anxiety, anger, loneliness, grief, and boredom [8]. Furthermore, these psychological stressors may lead to the development of serious mental health disorders [9,10].

During the early stages of the pandemic, different governments took unprecedented measures to safeguard the health of their citizens, including implementing a decrease in social contact and the isolation of large sectors of the population [11]. Anxiety, stress, fear, and phobia have been reported as predominant symptoms [1]. Anxiety during the pandemic may also be associated with several somatic symptoms, such as gastrointestinal sequelae and fatigue [12]. Some authors have characterized the COVID-19-related phobia (well known as “coronaphobia”) as an excessively triggered response of fear of contracting COVID-19, leading to excessive worry accompanied by physiological symptoms, significant stress from personal and occupational loss, increased safety and reassurance-seeking behaviours, and avoidance of public places and situations, causing a marked impairment in daily life functioning. This phobia was a response to the extreme concern of the population during the early stages of the pandemic [13].

After several months of quarantine, blockades, restrictions, and major repercussions on daily life worldwide, one of the consequences of the pandemic has been the generation of fatigue in the global population [14]. Pandemic fatigue is defined by the World Health Organisation as “demotivation to follow recommended protective behaviours, emerging gradually over time and affected by a number of emotions, experiences and perceptions” [15]. This effect is an expected and natural response to a sustained stressor among the general population [16]. Demotivation is an expected consequence after three years of a global pandemic: in the first stage people were able to draw on their coping capacities, a set of mental and physical adaptive systems adopted in the short term after acute stress; in the long term, the adoption of a different coping style leads to fatigue and demotivation, the so-called “pandemic fatigue” [15,16]. “Pandemic fatigue” should not be confused with fatigue as a part of the long-COVID syndrome: in this syndrome, patients describe persistent fatigue with the loss of energy, feelings of heaviness, and cognitive impairment (well known as “brain fog”) [17]. The fatigue is not relieved by rest and is accompanied by post-exertional malaise, unrefreshing sleep, cognitive impairment, or orthostatic intolerance [18].

The objective of this study was to assess the prevalence of “pandemic fatigue” among the adult population and to evaluate possible associated factors such as sociodemographic characteristics, health status, pandemic-related information, and the protective measures adopted. A large-scale evaluation was conducted based on a multi-center study involving Argentina, Bolivia, Uruguay, Peru, and Paraguay. For this purpose, we employed the Pandemic Fatigue Scale (PFS) developed and validated in 2021 in the context of the COVID-19 pandemic, which shows a bifactorial structure accounting for people’s demotivation in continuing to follow the recommended protective behaviours, and people’s boredom regarding the pandemic-related information [19].

## 2. Methods

### 2.1. Study Design

This was an observational cross-sectional study based on an online survey launched from the 1st of November to the 20th of December 2022. A total of 1448 respondents from Argentina, Bolivia, Uruguay, Peru, and Paraguay, of both sexes and aged ≥18 years old, voluntarily completed the survey, which was spread through common social media (“WhatsApp”, “Twitter”, and “Facebook”). All participants received complete information about the aim of the study, privacy, and data processing. No payment was offered for completing the survey. The study design is represented in Figure 1: we collected information on the country of origin, sex/gender, marital status, education, the mental health of participants, COVID-19 previous or current infection, hours spent using online media, and sources of news employed; a standardized assessment with The Pandemic Fatigue Scale (PFS) and The Coronavirus Anxiety Scale (CAS) was performed (as described below).

Figure 2 shows a diagram reporting the rate of responses (*y*-axis) received by date (*x*-axis).

### 2.2. Assessment Tools

The Pandemic Fatigue Scale (PFS), as developed and validated by Cuadrado et al., was employed in this study. The scale consists of a brief six-item questionnaire: three items assess the neglect factor (demotivation in continuing to follow the recommended protective measures), and three items assess the boredom factor (boredom regarding the pandemic-related information). Responses are provided using a Likert-type scale with answers ranging from 1 (“strongly disagree”) to 7 (“strongly agree”). This instrument was particularly suited for this study as it was developed on a Spanish-speaking population and was not influenced by gender. Higher scores are associated with an increased number of symptoms of pandemic fatigue [19].

The Coronavirus Anxiety Scale (CAS) [20] was employed in its Spanish version [21,22]. The CAS is a brief assessment instrument that measures physical responses to the stress related to the COVID-19 pandemic or coronaphobia. It consists of five questions (such as “I felt paralyzed or frozen when I thought about or was exposed to information about the coronavirus.”) with five possible answers, each on a scale ranging from 0–4. The scores of each question are added to produce the total CAS score ranging from 0–20. A higher score at the CAS indicates a greater level of physical reactions to coronaphobia. The CAS discriminates between subjects with and without dysfunctional anxiety using an optimized cut-off score ≥9 (90% sensitivity and 85% specificity). These results support the CAS as an efficient and valid tool for clinical research and practice [20].

### 2.3. Ethical Considerations

The present study was approved by the Department of Medical Psychology of the National University of Asunción (Paraguay; approval number 53/2022). Adherence to the principles of confidentiality, equality, and justice as outlined in the Helsinki Declaration were strictly maintained throughout the data collection and analysis process. Participants who requested feedback on their responses were invited to provide their email addresses and were subsequently informed of any relevant information or suggestion.

### 2.4. Statistical Analysis

All variables collected were recorded in a Microsoft Office Excel 2013 file and analysed using the RStudio statistical package version 1.2.5033. Results were presented in tables as proportions, and associations were evaluated using Student’s t-distribution and ANOVA, as appropriate. A *p*-value ≤ 0.05 was considered to indicate statistical significance.

## 3. Results

A total of 1448 participants with a mean age of 33.9 ± 11.2 years old and a median age of 32 years were included in the study. The Pandemic Fatigue Scale was employed in the study with a Cronbach’s alpha coefficient of 0.836. The scores obtained ranged from 6 to 42 points with a mean of 21.7± 7.95 and a median of 21 points. The boredom factor reported a Cronbach’s alpha of 0.825 with a mean of 10.4 ± 4.69, while the neglect factor reported a Cronbach’s alpha of 0.828 and a mean of 11.2 ± 4.54.

The majority of participants were from Paraguay, representing 20.9% of our total sample. With regards to gender, 72.4% of respondents were female. With regards to marital status, 49.5% of participants were single. A significant proportion of them (90.1%) had achieved a university education. The majority of participants (75%) were currently employed, and a significant proportion (17.4%) reported having lost their job during the pandemic. Additionally, 55.7% of participants reported experiencing economic losses due to the pandemic. Furthermore, 71.6% reported falling ill with COVID-19, whereas 59.3% reported having lost a family member or close friend during the pandemic. Women who lost a family member or friend due to the COVID-19 scored 20.93 ± 7.79 on the Pandemic Fatigue Scale, whereas men reported slightly higher scores: 21.90 ± 8.56. A significant proportion of participants (61.9%) had received two doses of the COVID-19 vaccine along with a booster. A slightly significant association between a personal history of COVID-19 infection and pandemic fatigue was found. Specifically, participants who had been diagnosed with COVID-19 exhibited more symptoms of fatigue than those who had not been diagnosed (21.9 ± 7.86 vs. 20.5 ± 8.09; F = 3.86, df = 2, *p* = 0.021). Those who had lost a relative or close friend during the pandemic demonstrated lower levels of pandemic fatigue compared to those who had not experienced such losses (21.1 ± 8.01 vs. 22.5 ± 7.79; *t*-test = 3.21, df = 1446, *p* = 0.001).

Participants who had received a vaccination reported fewer symptoms of pandemic fatigue compared to those who had not been vaccinated (22.0 ± 9.52 vs. 26.6 ± 8.85; F = 5.86, df = 4, *p* < 0.001). No association was found between pandemic fatigue scores and country of residence (Table 1). Furthermore, we found that 24.2% of responders had been diagnosed with a mental health disorder, 20.6% were currently under the care of a mental health professional, and 11.2% were regularly on psychotropic treatments. Nonetheless, no significant associations were found between these variables and pandemic fatigue. The most commonly used medications were antidepressants (72.8%), anxiolytics (53.1%), hypnotics (12.9%), antipsychotics (10.4%), and mood stabilizers (10.4%). A more detailed characterization of the sample is presented in Table 1.

The respondents identified work as the most significant source of stress in their lives (31.1%). Furthermore, 86.4% of participants reported spending 1–3 h per day reading information about COVID-19 in the previous month. The main sources of this information were social networks (70.6%; mostly Twitter: 64.1%). Compared to 4.92% of men, 5.24% of women spent seven or more hours a day on social networks. Of participants, 75.2% reported receiving information about COVID-19 from health or government agencies. These data are presented in further detail in Table 2. An analysis of these variables revealed a significant association between the source of information and pandemic fatigue, with greater pandemic fatigue among those gathering information from social networks (F = 3.99; df = 4; *p* = 0.003). Furthermore, among social-media platforms, Instagram was strongly associated with a higher level of pandemic fatigue (F = 5.80, df = 4; *p* < 0.001). Additionally, participants who received information from friends reported a high level of fatigue, whereas those who received information from government agencies reported lower levels (F = 5.90; df = 4; *p* = 0.001; Table 2).

Figure 3 illustrates the frequency of the utilization of non-pharmacological protection measures against COVID-19. The most commonly adopted measures were hand washing, the use of hand sanitizer, and face mask usage in enclosed spaces. It was observed that the consistent utilization of these protective measures was positively correlated with a reduction in symptoms of pandemic fatigue, as evidenced by a statistically significant association (Table 3).

In regard to anxiety related to the coronavirus, the scores reported that the anxiety scale ranged from 0 to 20 points with a mean of 1.56 ± 2.76 and a Cronbach’s alpha of 0.874. According to the established cut-off points, 3.9% of participants presented significant anxiety related to COVID-19. A slightly significant correlation was found between participants’ anxiety and pandemic fatigue score (*r* = 0.069; *p* = 0.008).

## 4. Discussion

This study confirmed that pandemic fatigue is now acknowledged as a prevalent response to the prolonged challenges of the COVID-19 pandemic and the measures implemented by nations worldwide to contrast it among the general population [16,23]. Our findings reported a mean score on the Pandemic Fatigue Scale of 21.7 ± 7.95 (boredom factor: 10.4 ± 4.69; neglect factor: 11.2 ± 4.54). These scores were similar to those obtained in a research study conducted in Spain (PFS total score: 17.06 ± 5.04) and Saudi Arabia (PFS total score: 17.8 ± 7.0) [24,25].

Although we have not found a significant association between fatigue levels and gender, recent research studies have found that women were more likely to suffer from fatigue since they had to deal with additional tasks during the pandemic such as children’s home-schooling and more domestic work [26]. Similarly, lower levels of education were significantly associated with higher pandemic fatigue in some research findings [27,28]; we did not find a significant association between levels of education and levels of fatigue across the surveyed countries.

Notably, a significant association was observed between the experience of a COVID-19-related loss of a close friend or relative and higher levels of pandemic fatigue. This finding has been confirmed in several studies reporting that the experience of loss related to COVID-19 led to severe consequences in terms of anxiety, stress, and depression; higher levels of these symptoms were also associated with more perceived fatigue [29,30,31]. In addition, the loss of a loved one is widely acknowledged as a traumatic event that significantly increases pandemic fatigue [32].

We found a significant association between pandemic fatigue and a previously confirmed diagnosis of COVID-19 infection. Some other studies confirmed that patients that recovered from the infection reported exhaustion, lack of motivation, and isolation due to pandemic fatigue in the following months [33]. In addition, our findings indicated that vaccination (two doses plus one/two boosters) was associated with lower pandemic fatigue as confirmed in a survey of 255 frontline clinical nurses from the Philippines [34]. Unexpectedly, no association was observed between pandemic fatigue and a diagnosis of mental health disorders; this result may be biased by the low frequency of anxiety symptoms related to COVID-19 in the sample.

According to our findings, gathering information about the pandemic via social media, particularly Instagram, was associated with significantly higher levels of pandemic fatigue. A similar survey conducted on 849 social media users in China confirmed that the amount of information, or “content overload”, significantly contributed to pandemic fatigue [35]. Additionally, people reported increased use of media to fill the gap in their social lives, especially during the first stage global lockdown, with consequences in terms of isolation and mental health issues such as depression, anxiety, and low self-esteem [36]. The higher percentage of female respondents in this survey may suggest their higher use of media in South America; these are misleading conclusions since this survey was conducted at the end of the year 2022 and is not connected to the substitute employment of online media for coping with restrictions and isolation due to the first stage of global lockdown. In addition, women did not report higher levels of pandemic fatigue than males as already discussed. In general, it has been described as a greater attitude of women to take part in e-mail surveys mostly based on their personality characteristics [37].

Non-pharmacological protective measures were essential for contrasting the transmission of COVID-19 infections. These measures limited physical contact with others and led to regular hand washing and facemasks employment; it has been argued that by implementing these non-pharmacological protective measures, individuals might reduce the pandemic fatigue levels by reducing the stress of constantly worrying about the potential risk of infection [38]. Our findings confirmed that higher adherence to prevention measures was associated with fewer symptoms of pandemic fatigue.

Finally, a slightly significant positive correlation was found between anxiety related to COVID-19 and pandemic fatigue, which is consistent with previous studies [39]. Previous studies on the psychological impacts of the pandemic have suggested that feelings of stress and anxiety related to the virus were common [40].

The limitations of this study may include the sampling through social networks, a possible self-selection bias among respondents, and an overrepresentation of women, young people, and participants with higher levels of education. In general, people with higher levels of education are more likely to participate in the surveys [41], and women are more inclined to participate than men and youths more than elderly people [36].

Strengths of this study include the employment of highly specific and validated tools with appropriate psychometric properties and a large sampling with a multi-center design involving the general population from five representative countries of South America.

## 5. Conclusions and Final Remarks

In conclusion, we found mid-levels of pandemic fatigue in South America. Significant associations were found between pandemic fatigue and the loss of a relative/friend due to COVID-19, anxiety related to the infection, and reliance on social media as a primary source of information on the pandemic. Vaccination significantly reduced the levels of fatigue among respondents. Our findings may add to the international debate regarding the long-term health consequences of the COVID-19 pandemic and strategies to manage them in the general population of South America. In particular, plans based on social resilience should be adopted at a governmental level; positive messages should be spread through the general population on the current overcoming of the pandemic, the effectiveness of vaccination campaigns, and information about the lower clinical severity of COVID-19 infection. In addition, the easing of restrictions and safety measures should lead to a new boost of social, cultural, and economic activities in the framework of a global recovery package. Mental health services should provide specific support for those people suffering from grief due the loss of relatives and friends because of COVID-19 as well those with as a personal history of severe life-threatening infection. Governmental policies and mental health specific interventions may reduce the long-term effects of pandemic, including the pandemic fatigue.

## Figures and Tables

**Figure 1 brainsci-13-00444-f001:**
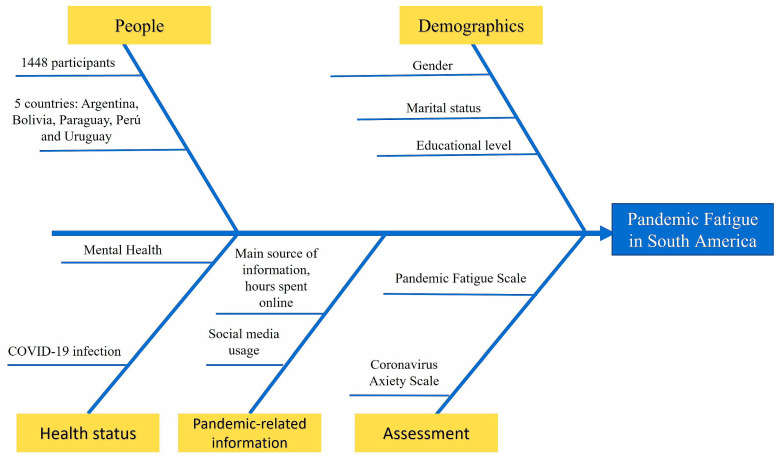
Ishikawa (fishbone) diagram of the methods.

**Figure 2 brainsci-13-00444-f002:**
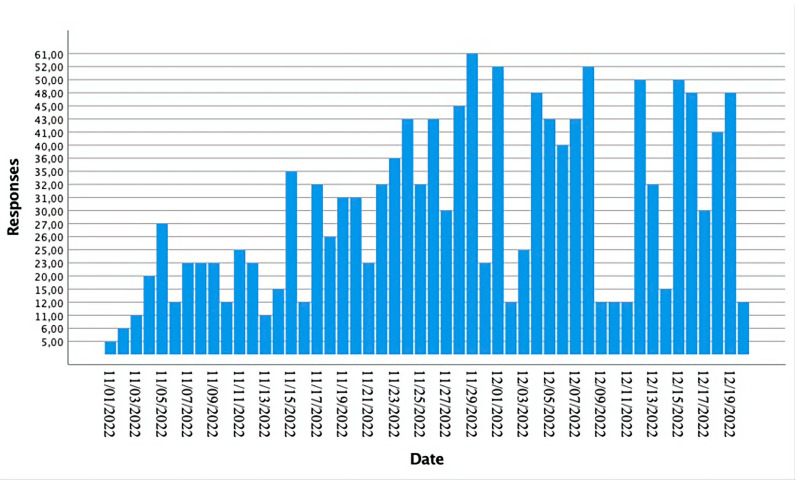
Survey responses received by date (N = 1448).

**Figure 3 brainsci-13-00444-f003:**
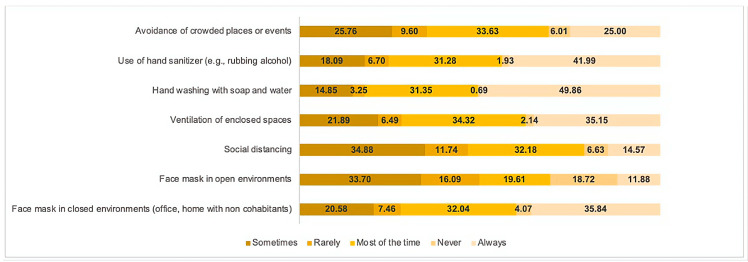
Non-pharmacological protective measures against COVID-19 in South America (N = 1448).

**Table 1 brainsci-13-00444-t001:** Associated characteristics to the Pandemic Fatigue Scale (PFS scores) among respondents from South America (N = 1448).

Characteristics	n	%	Mean	SD	SE	*p*-Value
**Country** Argentina Bolivia Uruguay Peru Paraguay **Gender** Female Male Non-binary I prefer not to say **Marital status** Partnered—married Separated—divorced Single Widowed **Education** Primary education Secondary education University education **Previous infection by COVID-19** Not I don’t know Yes **Loss of a relative or close friend during the pandemic** Yes Not **Vaccination** I have not been vaccinated Yes, two doses Yes, two doses plus two boosters Yes, two doses plus one booster Yes, one dose **Diagnosed with a mental disorder** Not Yes **Currently under care of mental health professionals** Not Yes **Regularly on psychotropic treatments** Not Yes	295 294 279 277 303 1048 386 6 8 645 71 717 15 2 142 1304 266 145 1037 859 589 23 229 284 897 15 1149 299 1097 351 1286 162	20.4 20.3 19.3 19.1 20.9 72.4 26.7 0.4 0.6 44.5 4.9 49.5 1.0 0.1 9.8 90.1 18.4 10.0 71.6 59.3 40.7 1.6 15.8 19.6 61.9 1.0 79.4 20.6 75.8 24.2 88.8 11.2	21.62 21.89 21.14 22.60 21.47 21.62 21.99 22.00 24.38 21.56 19.52 22.11 22.07 17.50 20.99 21.83 20.53 22.24 21.98 22.54 21.18 26.65 23.33 20.75 21.51 22.00 21.55 22.46 21.56 22.29 21.68 22.16	7.77 7.95 8.01 8.15 7.85 7.74 8.42 12.92 8.25 7.88 7.98 7.95 9.15 3.54 8.86 7.84 8.09 8.16 7.86 7.79 8.01 8.85 8.24 8.62 7.50 9.52 7.95 7.91 7.97 7.87 7.92 8.14	0.45 0.46 0.48 0.49 0.45 0.24 0.43 5.27 2.92 0.31 0.95 0.29 2.36 2.50 0.74 0.22 0.50 0.68 0.24 0.32 0.27 1.84 0.54 0.51 0.25 2.46 0.23 0.46 0.24 0.42 0.22 0.64	0.255 0.681 0.06 0.369 0.021 0.001 <0.001 0.078 0.134 0.472

PFS: the Pandemic Fatigue Scale; SD: standard deviation; SE: standard error.

**Table 2 brainsci-13-00444-t002:** Sources of information on the COVID-19 pandemic and associations with levels on the Pandemic Fatigue Scale (PFS scores) (N = 1448).

Characteristics	n	%	Mean	SD	SE	*p*-Value
**Major source of stress** Money Study None Intimate/Family Relationships Work Housing **Hours spent in information ** 1 to 3 h per day 4 to 6 h per day 7 to 8 h per day More than 8 h per day **Main source of information** Newspapers Radio Social media Scientific journals TV **Social media** Facebook Instagram Tik-Tok Twitter **Main source of information** Friends Coworkers Family Health/government agencies	418 204 75 266 451 34 1251 121 28 48 65 43 1022 97 221 240 107 19 656 107 91 161 1089	28.9 14.1 5.2 18.4 31.1 23 86.4 8.4 1.9 3.3 4.5 3.0 70.6 6.7 15.3 23.48 10.47 1.86 64.19 7.4 6.3 11.1 75.2	21.85 21.80 19.95 22.60 21.42 21.35 21.75 20.65 22.11 24.02 19.92 19.95 22.21 19.77 21.29 21.39 23.50 18.26 22.42 23.84 22.84 23.02 21.25	8.32 7.97 7.36 7.65 7.95 5.91 7.92 8.24 7.21 8.07 6.37 7.06 7.86 8.50 8.45 7.68 8.28 5.14 7.87 8.81 7.91 7.71 7.84	0.41 0.56 0.85 0.47 0.37 1.01 0.22 0.75 1.36 1.16 0.79 1.08 0.25 0.86 0.57 0.50 0.80 1.18 0.31 0.85 0.83 0.61 0.24	0.166 0.099 0.003 <0.001 0.001

PFS: the Pandemic Fatigue Scale; SD: standard deviation; SE: standard error.

**Table 3 brainsci-13-00444-t003:** Non-pharmacological protective measures against COVID-19 and association with the Pandemic Fatigue Scale (PFS scores) (N = 1448).

Protective Measures	Mean	SD	SE	*p*-Value
**Face mask in closed environments** Sometimes Rarely Most of the time Never Always **Face mask in open environments** Sometimes Rarely Most of the time Never Always **Social distancing** Sometimes Rarely Most of the time Never Always **Ventilation of enclosed spaces** Sometimes Rarely Most of the time Never Always **Hand washing** Sometimes Rarely Most of the time Never Always **Use of hand sanitizer** Sometimes Rarely Most of the time Never Always **Avoidance of crowded places or events** Sometimes Rarely Most of the time Never Always	24.57 25.93 21.27 28.39 18.90 21.61 22.81 18.82 26.25 18.34 22.12 24.80 20.32 27.44 18.90 23.91 23.28 21.35 29.00 20.04 23.81 24.68 22.46 26.20 20.41 23.54 24.91 21.84 27.57 20.11 23.13 24.06 20.44 28.33 19.56	7.82 8.42 6.98 7.60 7.42 7.60 7.24 6.63 8.09 7.94 7.77 7.35 7.10 8.89 8.02 8.04 6.99 7.37 8.06 8.01 8.35 7.47 7.72 7.24 7.76 8.08 6.95 7.46 9.14 7.90 7.50 7.38 7.24 8.25 8.12	0.45 0.81 0.32 0.99 0.32 0.34 0.47 0.39 0.49 0.61 0.35 0.56 0.33 0.91 0.55 0.45 0.72 0.33 1.45 0.36 0.57 1.09 0.36 2.29 0.29 0.50 0.71 0.35 1.73 0.32 0.39 0.63 0.33 0.88 0.43	<0.001 <0.001 <0.001 <0.001 <0.001 <0.001 <0.001

PFS: the Pandemic Fatigue Scale; SD: standard deviation; SE: standard error.

## Data Availability

The data that support the findings of this study are available from the corresponding author upon reasonable request.

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
