# Peer review of "“Pandemic Fatigue” in South America: A Multi-Center Report from Argentina, Bolivia, Paraguay, Peru, and Uruguay"

_brainsci, 2023, doi:10.3390/brainsci13030444_

Round 1

Reviewer 1 Report

Introduction:

1.      Elaborate more about how you differentiate long term COVID-19 syndrome, pandemic fatigue and corona phobia. One paragraph with 7-8 lines completely dedicated to this in introduction.

Methods.

1.      Represent your study design using fishbone diagram.

2.      This survey involved participants from males (n= 386) and females (n=1048), looks like more females responded to this survey than males. Why such discrepancies in the beginning itself? I know the author doesn’t have any control over it. Still one can imagine the women were more likely to suffer from fatigue since they had to deal with additional tasks during the pandemic such as children’s home-schooling and more domestic work than male counter part as author mentioned in the discussion part. One thing to be noted, does women spend more time on social media than men?

3.      The author mentioned an online survey launched from 1st of November to 20th of December 2022 and total 1448 respondents. The author must represent this data in one single bar graph, showing on Y-axis number of days of survey Vs X-axis people responded per day. Just to see how people responded in the beginning, middle and later.

Statistics.

1.      Author used standard deviation (SD) to analyze data but this SD is more common to express spread and variability of any data. Author must recheck their data or parameters which showing <0.001 significant by comparing or testing the precision of the means using standard errors (SE).

2.      The author found strong significant associations between pandemic fatigue and the loss of a relative/friend due to COVID-19. Author must add details about who suffered most of the emotional damage, is it a male or female?

Discussion.

1.      In discussion author mentioned that an increased use of media to fill the gap in their social lives with consequences in terms of isolation and mental health issues such as depression, anxiety and low self-esteem. Doesn’t fit well with data sets of this survey as having high number of females ended up attending this survey. That shows only women started using social media to fill the gap in their social lives. Also showing they are going through more mental health issues than males.

That means this survey become one-sided or inclined towards one sex that is females.

Please defend this in your discussion part.

Author Response

Responses to REVIEWER #1

Introduction:

  1. Elaborate more about how you differentiate long term COVID-19 syndrome, pandemic fatigue and corona phobia. One paragraph with 7-8 lines completely dedicated to this in introduction.

Response: An extensive paragraph has been added. Many thanks for this suggestion.

Methods.

  1. Represent your study design using fishbone diagram.

Response: A fishbone diagram (new Figure 1) has been added in the methods section.

  1. This survey involved participants from males (n= 386) and females (n=1048), looks like more females responded to this survey than males. Why such discrepancies in the beginning itself? I know the author doesn’t have any control over it. Still one can imagine the women were more likely to suffer from fatigue since they had to deal with additional tasks during the pandemic such as children’s home-schooling and more domestic work than male counter part as author mentioned in the discussion part. One thing to be noted, does women spend more time on social media than men?

Response: The overrepresentation of women has been noted in the discussion (among limitations) section of the manuscript. Some references from the literature have been cited regarding the attitudes to surveys in women, youths and people with higher levels of education.

Regarding the question about social media usage, we have reported time spent on social media in the results section.

  1. The author mentioned an online survey launched from 1stof November to 20th of December 2022 and total 1448 respondents. The author must represent this data in one single bar graph, showing on Y-axis number of days of survey Vs X-axis people responded per day. Just to see how people responded in the beginning, middle and later.

Response: Many thanks for this helpful suggestion. We have included a graph (as new Figure 2), with the date on X-axis and responses in Y-axis, for more clarity.

Statistics.

  1. Author used standard deviation (SD) to analyze data but this SD is more common to express spread and variability of any data. Author must recheck their data or parameters which showing <0.001 significant by comparing or testing the precision of the means using standard errors (SE).

Response: Many thanks for this suggestion. We have included a SE column in all tables.

  1. The author found strong significant associations between pandemic fatigue and the loss of a relative/friend due to COVID-19. Author must add details about who suffered most of the emotional damage, is it a male or female?

Response: Many thanks for this suggestion. we have added information about the mean score on the pandemic fatigue among sex groups in the results section.

Discussion

  1. In discussion author mentioned that an increased use of media to fill the gap in their social lives with consequences in terms of isolation and mental health issues such as depression, anxiety and low self-esteem. Doesn’t fit well with data sets of this survey as having high number of females ended up attending this survey. That shows only women started using social media to fill the gap in their social lives. Also showing they are going through more mental health issues than males. That means this survey become one-sided or inclined towards one sex that is females. Please defend this in your discussion part.

Response: This is a good point. We discussed the topic as following (a paragraph has been added in the discussion):

Additionally, people reported an increased use of media to fill the gap in their social lives, especially during the first stage global lockdown, with consequences in terms of isolation and mental health issues such as depression, anxiety and low self-esteem [36]. The higher percentage of female respondents in this survey may suggest their higher use of media in South America: these are misleading conclusions since this survey has been conducted in the end of year 2022 and is not connected to the substitute employment of online media for coping with restrictions and isolation due to the first stage- global lockdown. Also,  women did not report higher levels of pandemic fatigue than males as already discussed. In general, it has been described a greater attitude of women to take part into e-mail surveys mostly based on their personality characteristics [37].      

Reviewer 2 Report

The authors of this paper reported interesting findings of a large survey involving 1448 participants from Argentina, Bolivia, Uruguay, Peru and Paraguay. They found mid-levels of pandemic fatigue among respondents as well as significant anxiety related to COVID-19 pandemic. The objective of the study was to assess the prevalence of “pandemic fatigue” among the adult population and to evaluate possible associated factors including sociodemographic characteristics, health status, pandemic-related information, and protective measures. For this purpose, the authors employed the Pandemic Fatigue Scale (PFS), which shows a bifactorial structure accounting for people’s demotivation in continuing to follow the recommended protective behaviors, and people’s boredom regarding the pandemic-related information.

The literature presented in the references is relevant and up-to-date and the sources are rich. And, which is rather rare in the papers I used to review, the references are prepared almost perfectly. I can only indicate two minor technical errors:

– In item 11, the name of the journal should be written as: PLoS ONE (instead of: PloS One) – such a notation is suggested by the editors of the journal themselves;

– In item 25, prepositions in the title of the paper should be written with a lowercase letter, i.e.: ‘with’, ‘among’, and ‘during’.

Generally, the paper should be verified from the editorial (technical) point of view. Authors’ English is correct, however it requires linguistic proofreading.

In the case of the Introduction section, it would not hurt to supplement it with a few additional sources, because although there is a number of them, it is worth squeezing a bit more at this point. On the other hand, the Conclusion segment should be a bit more developed because it is too laconic. Perhaps it should be supplemented with specific proposals for further research in the undertaken area. Or the solution would be including the last paragraph from the Discussion section in the Conclusion segment and converting the heading into Limitations and Conclusions. These are just two suggestions.

Author Response

Responses to REVIEWER #2

  1. The authors of this paper reported interesting findings of a large survey involving 1448 participants from Argentina, Bolivia, Uruguay, Peru and Paraguay. They found mid-levels of pandemic fatigue among respondents as well as significant anxiety related to COVID-19 pandemic. The objective of the study was to assess the prevalence of “pandemic fatigue” among the adult population and to evaluate possible associated factors including sociodemographic characteristics, health status, pandemic-related information, and protective measures. For this purpose, the authors employed the Pandemic Fatigue Scale (PFS), which shows a bifactorial structure accounting for people’s demotivation in continuing to follow the recommended protective behaviors, and people’s boredom regarding the pandemic-related information. The literature presented in the references is relevant and up-to-date and the sources are rich. And, which is rather rare in the papers I used to review, the references are prepared almost perfectly. I can only indicate two minor technical errors: 
  • In item 11, the name of the journal should be written as: PLoS ONE (instead of: PloS One) – such a notation is suggested by the editors of the journal themselves;
  • In item 25, prepositions in the title of the paper should be written with a lowercase letter, i.e.: ‘with’, ‘among’, and ‘during’.

Response: Many thanks for these suggestions. We have corrected references 11 and 25 (now references 14 and 30, respectively).

  1. Generally, the paper should be verified from the editorial (technical) point of view. Authors’ English is correct, however it requires linguistic proofreading.

Response: we have revised the manuscript accordingly

  1. In the case of the Introduction section, it would not hurt to supplement it with a few additional sources, because although there is a number of them, it is worth squeezing a bit more at this point.

Response: Many thanks. We have improved the Introduction section above all reporting more description of the long-COVID syndrome, fatigue and coronaphobia, which are relevant topics for the international scientific debate.

  1. On the other hand, the Conclusion segment should be a bit more developed because it is too laconic. Perhaps it should be supplemented with specific proposals for further research in the undertaken area. Or the solution would be including the last paragraph from the Discussion section in the Conclusion segment and converting the heading into Limitations and Conclusions. These are just two suggestions.

Response: this is a good point. We developed the Conclusion and switched into “Conclusions and final remarks”. This paragraph has been added

In particular, plans based on social resilience should be adopted at governmental level: positive messages should be spread through the general population on the current overcoming the pandemic, the effectiveness of vaccination campaigns  and information about the lower clinical severity of COVID-19 infection. Also, the easing of restrictions and safety measures should lead to a new boost of social, cultural and economic activities in the framework of a global recovery package. Mental health services should provide a specific support for those people suffering from the grief due the loss of relatives and friends because of COVID-19 as well as a personal history of severe, life-threatening infection. Governmental policies and mental health specific interventions may reduce the long-term effects of pandemic, including the pandemic fatigue.

Responses to REVIEWER #2

  1. The authors of this paper reported interesting findings of a large survey involving 1448 participants from Argentina, Bolivia, Uruguay, Peru and Paraguay. They found mid-levels of pandemic fatigue among respondents as well as significant anxiety related to COVID-19 pandemic. The objective of the study was to assess the prevalence of “pandemic fatigue” among the adult population and to evaluate possible associated factors including sociodemographic characteristics, health status, pandemic-related information, and protective measures. For this purpose, the authors employed the Pandemic Fatigue Scale (PFS), which shows a bifactorial structure accounting for people’s demotivation in continuing to follow the recommended protective behaviors, and people’s boredom regarding the pandemic-related information. The literature presented in the references is relevant and up-to-date and the sources are rich. And, which is rather rare in the papers I used to review, the references are prepared almost perfectly. I can only indicate two minor technical errors:
  • In item 11, the name of the journal should be written as: PLoS ONE (instead of: PloS One) – such a notation is suggested by the editors of the journal themselves;
  • In item 25, prepositions in the title of the paper should be written with a lowercase letter, i.e.: ‘with’, ‘among’, and ‘during’.

Response: Many thanks for these suggestions. We have corrected references 11 and 25 (now references 14 and 30, respectively).

  1. Generally, the paper should be verified from the editorial (technical) point of view. Authors’ English is correct, however it requires linguistic proofreading.

Response: we have revised the manuscript accordingly

  1. In the case of the Introduction section, it would not hurt to supplement it with a few additional sources, because although there is a number of them, it is worth squeezing a bit more at this point.

Response: Many thanks. We have improved the Introduction section above all reporting more description of the long-COVID syndrome, fatigue and coronaphobia, which are relevant topics for the international scientific debate.

  1. On the other hand, the Conclusion segment should be a bit more developed because it is too laconic. Perhaps it should be supplemented with specific proposals for further research in the undertaken area. Or the solution would be including the last paragraph from the Discussion section in the Conclusion segment and converting the heading into Limitations and Conclusions. These are just two suggestions.

Response: this is a good point. We developed the Conclusion and switched into “Conclusions and final remarks”. This paragraph has been added

In particular, plans based on social resilience should be adopted at governmental level: positive messages should be spread through the general population on the current overcoming the pandemic, the effectiveness of vaccination campaigns  and information about the lower clinical severity of COVID-19 infection. Also, the easing of restrictions and safety measures should lead to a new boost of social, cultural and economic activities in the framework of a global recovery package. Mental health services should provide a specific support for those people suffering from the grief due the loss of relatives and friends because of COVID-19 as well as a personal history of severe, life-threatening infection. Governmental policies and mental health specific interventions may reduce the long-term effects of pandemic, including the pandemic fatigue.

Reviewer 3 Report

The long-term effects of COVID-19, is a topic of great interest to many researchers. The observed phenomenon of 'long COVID-19 fatigue' is observed in many countries. In this paper, the authors present the results of a large study conducted in South America and involving 1,448 participants This was a web-based study. The authors corroborated reports from other countries and found significant associations between pandemic fatigue and loss of a relative/friend due to COVID-19, infection-related anxiety and reliance on social media as the main source of information about the pandemic. Also important is the result indicating an association of vaccination with significantly reduced levels of COVID-19 fatigue. The paper is interesting. The authors should work on the editorial side e.g. Table 1 in its entirety should be moved to the next page, currently the table header is on a different page than the content. 

Author Response

Responses to REVIEWER #3

  1. The long-term effects of COVID-19, is a topic of great interest to many researchers. The observed phenomenon of ‘long COVID-19 fatigue’ is observed in many countries. In this paper, the authors present the results of a large study conducted in South America and involving 1,448 participants This was a web-based study. The authors corroborated reports from other countries and found significant associations between pandemic fatigue and loss of a relative/friend due to COVID-19, infection-related anxiety and reliance on social media as the main source of information about the pandemic. Also important is the result indicating an association of vaccination with significantly reduced levels of COVID-19 fatigue. The paper is interesting. The authors should work on the editorial side e.g. Table 1 in its entirety should be moved to the next page, currently the table header is on a different page than the content. 

Response: Many thanks for this suggestion. We have corrected the position of table 1.
